# Study on Purification Efficiency of Novel Aquatic Plant Combinations and Characteristics of Microbial Community Disturbance in Eutrophic Water Bodies

**Jianna Jia [1], Huan Xiao [2], Shitao Peng [1,\*] and Kailei Zhang [1]**

[1] Tianjin Research Institute for Water Transport Engineering, Ministry of Transport, Tianjin 300456, China; jiajn@tiwte.ac.cn (J.J.); zhangkl@tiwte.ac.cn (K.Z.)
[2] School of Environment and Ecology, Chongqing University, Chongqing 400044, China; xh5160@foxmail.com
\* Correspondence: pengshitaotj@163.com

**Abstract:** Aquatic plant restoration is an important technique for the treatment of eutrophic water bodies. There are significant differences in pollutant removal efficiency among different combinations of aquatic plant species in eutrophic water bodies. Therefore, further research on the selection of suitable combinations of aquatic plant species is of great significance for the restoration of eutrophic water bodies. This study investigated the pollutant removal efficiency and bacterial community structure of three novel combinations of aquatic plants, including Lythraceae, Nymphaea, and Myriophyllum (LNM group), Lythraceae, Nymphaea, and Hydrilla verticillata (LNH group), and Lythraceae, Nymphaea, and Vallisneria (LNV group), as well as a control group (CK group). The components of the CK group were only sediment and culture water without any plants. The results show that on one hand, the LNH group had the highest removal rate of COD (90.29%); the LNV group exhibited the highest removal rates for $NH_4^+$-N and TN, with removal rates of 61.20% and 82.94%, respectively; and there was no significant difference in the removal rate of TP among the experimental groups, except for the LNH group, which showed higher initial removal efficiency for TP. On the other hand, plant combinations had different impacts on the top 13 dominant microflora at the phylum level. Proteobacteria and Actinobacteria showed the highest removal efficiency for COD in the LNH group, while Verrucomicrobi, Chloroflex, and Acidobacteria showed higher removal efficiency for $NH_4^+$-N and TN in the LNV and LNH groups. In summary, the three different combinations of aquatic plants exhibited distinct pollutant removal characteristics, significantly altered the structure of the microbial community, and provided a theoretical basis for their practical application in the restoration of eutrophic water bodies.

**Keywords:** eutrophication; pollutant removal; phytoremediation; microbial communities

## 1. Introduction

The accumulation of nutrients and eutrophication is a threat to most water bodies worldwide [1]. Eutrophication has become one of the most serious water pollution problems in Chinese lakes and is prevalent in various water bodies globally. Eutrophication has profound negative impacts on surface water quality and ecological environment globally and poses a hidden danger to biological and human health [2,3]. Eutrophication of water bodies can trigger algal blooms and water blooms, deplete oxygen in the water, cause water turbidity, damage fishery resources, affect drinking water safety, and disrupt ecological balance, among a series of environmental issues [4]. Numerous restoration techniques have been employed in eutrophication control and can be broadly categorized into four types: physical methods, chemical methods, biological restoration, and comprehensive restoration technologies [5]. Among all restoration techniques, biological restoration has been favored by scholars due to its excellent effectiveness and lower economic costs. Biological restoration mainly reduces nutrient levels in water bodies through absorption and metabolism of

nitrogen and phosphorus by aquatic plants, animals, and microorganisms. It is known as a simple, environmentally friendly, and widely applicable method [6,7].

Aquatic plants, including submerged, emergent, and floating plants, play a vital role in purifying water bodies by absorbing, transferring, and transforming nitrogen, phosphorus, and other organic matter from water and sediment into their own biomass [8–10]. Among them, submerged plants directly assimilate nutrients from water, while emergent plants assimilate nutrients through their roots. The removal efficiencies of nutrients by emergent, floating, and submerged plants were approximately 5%, 25%, and 40%, respectively [8]. In recent years, extensive studies have been conducted on the removal of pollutants by various aquatic plants, such as water hyacinth (*Eichhornia crassipes*), duckweed (*Lemna* spp.), water lettuce (*Pistia stratiotes*), vetiver grass (*Chrysopogon zizanioides*), common reed (*Phragmites australis*), and *Lythrum salicaria* L. [11,12]. Furthermore, studies have shown that a combination of different plant species may have a better water purification effect than a single species. For example, Xu et al. investigated the performance of *Phragmites australis*, *Nymphaea alba*, and *Myriophyllum verticillatum* and their combination on the restoration of eutrophic water containing chlorpyrifos [13]. The removal rates of COD, TN, and TP were generally in the following order: plant combination > *Myriophyllum verticillatum* > *Nymphaea alba* > *Phragmites australis*. Xiao et al. constructed a plant combination system consisting of Lythraceae, Nymphaea, and Vallisneria, which achieved a COD removal rate of 92.4% [14]. Aquatic plants are an important component of bioremediation technology, but different species of aquatic plants have different pollutant removal effects and mechanisms. Therefore, there is a promising potential for improving pollutant removal by screening different species of plants. However, the best combination of emergent, floating, and submerged plants for optimal pollution removal have not been thoroughly studied yet.

In addition, it is widely recognized that microbes also play a significant role in pollutant removal through organic nitrogen degradation, nitrification, denitrification, stabilization, volatilization, and other processes [15,16]. There is a complex interaction between plants and microorganisms, where plants provide an aerobic environment and organic compounds for the growth and reproduction of microorganisms [17,18]. At the same time, plants can directly or indirectly affect the biomass and productivity of planktonic bacteria by absorbing essential elements for life. Zuo et al. found that Ramlibacter and Nitrosomonadaceaea were dominant microflora for nitrogen removal in bioretention cells planted with *Iris pseudacorus* L., *Canna indica* L., and *Lythrum salicaria* L [19]. Cheng et al. investigated the microbial community structure in constructed wetlands (CWs) planted with giant reed under different substrate conditions and found that the relative abundance of Proteobacteria was the highest in CWs using iron–carbon as a substrate, while the highest relative abundances of Comamonadaceae, Planctomycetaceae, and Burkholderiaceae were observed in CWs using ceramsite as a substrate [20]. Therefore, different types of plants altered microbial abundance or activity [21,22]. Therefore, in this study, the performances of pollutant removal in a bioremediation system with different plant combinations were investigated. The microbial diversity of different plant combinations was analyzed, as well as the effect of microbial diversity on pollutant removal, for the improvement of pollutant removal.

Overall, the effects of different plant combinations on pollutant removal have not been well evaluated, and the response of microorganisms to different plant combinations is also unknown. Therefore, the objectives of this study were as follows: (1) to evaluate the removal efficiency of nutrients from eutrophic water under different plant combinations in order to obtain the optimal plant combination for bioremediation of eutrophic water; (2) to reveal the response of the microbial community to the presence of different plants in order to identify dominant microbial communities that can further enhance the pollutant removal efficiency of plant combinations. This study investigates the interactions among different aquatic plants to uncover the synergistic effects of various combinations of aquatic plants in the restoration of eutrophicated water bodies, which establishes a comprehensive restoration strategy using a combination of aquatic plants for eutrophication control in water bodies. This study will

provide a theoretical basis for the practical engineering application of the combined plant system by submerged, emergent, and floating plants in eutrophication control.

## 2. Materials and Methods

### 2.1. Preparation of Plant Materials and Pot Experimental Design

Previous research has shown that compared with emergent and floating plants, submerged plants play a more critical role in water purification. Therefore, for emergent and floating plants, we selected species with simultaneous landscape effects and water purification effects, which were Lythraceae and Nymphaea. And for the accompanying submerged plants, we chose species that have been widely used in previous research with excellent water purification effects, which were Myriophyllum, Hydrilla verticillata, and Vallisneria, respectively. Among them, it has been demonstrated through research that Myriophyllum, as a submerged aquatic plant, has significant effects on the removal rates of COD, TN, and TP in eutrophic water bodies [13,14]. The species names of Lythraceae, Myriophyllum, and Vallisneria were *Lythrum salicaria* L., *Myriophyllum spicatum* L., and *Vallisneria natans* (Lour.) Hara in this study, respectively. To investigate the optimal plant combination for water purification, we constructed three combined plant systems by submerged, emergent, and floating plants as follows, and a blank group was designed as a control group:

(1)　LNM group: Lythraceae, Nymphaea, and Myriophyllum;
(2)　LNH group: Lythraceae, Nymphaea, and Hydrilla verticillata;
(3)　LNV group: Lythraceae, Nymphaea, and Vallisneria;
(4)　CK group: contained only sediment and culture water without any plants.

Healthy young plants were obtained from Honghu Lvshui Aquaculture Base (Hubei, China), and sediments were collected in situ. The collected sediments were filtered through a sieve (20 mesh, diameter 0.85 mm) and washed with tap water. The plants were then cultured for 14 days in a plastic water tank (160 L; 80 cm × 40 cm × 50 cm) with a sediment depth of 60 mm and a water depth of 35 cm under good lighting conditions indoors at the Tianjin Research Institute for Water Transport Engineering (117°41′53″ E, 39°0′11″ N), Tianjin, China (Figure 1). After the culture period, healthy young plants were selected for the experiment. Prior to the experiment, all plants were washed with clean water. The length of Lythraceae, Nymphaea, and submerged plants were cut to 70 cm, 40 cm, and 20 cm, respectively. And the planting densities of Lythraceae, Nymphaea, and submerged plants were 10 plants/m$^2$, 50 clumps/m$^2$, and 25 plants/m$^2$, respectively. The plants were spaced at a fixed distance in each water tank. Three water tanks were set up as parallel samples for each of the three combined plant systems (LNM, LNH, and LNV) and the control group (CK). The 12 tanks were arranged side by side in a laboratory with good lighting and stable temperature. During the experiment, the water level was maintained at 35 cm by adding water to the tanks every three days. The culture water was prepared by mixing glucose, $NH_4Cl$, $KNO_3$, and $KH_2PO_4$. The influent water quality was maintained at a certain level during the whole experiment (Table 1).

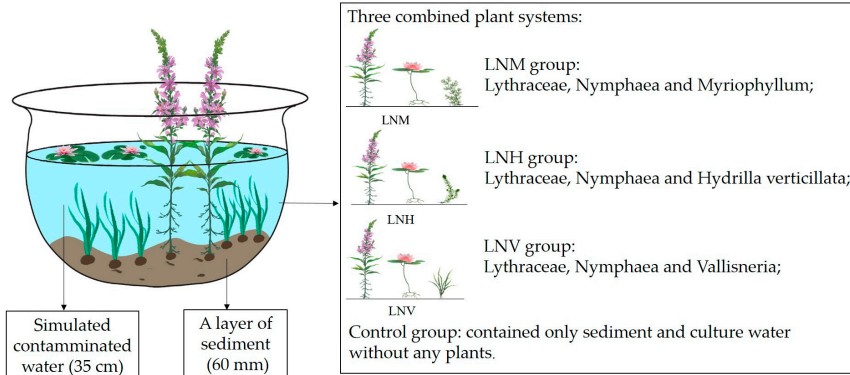

**Figure 1.** Schematic of constructed combined plants system for purification of eutrophic water.

**Table 1.** Water quality of the synthetic eutrophic water.

| COD (mg/L) | NH$_4$-N$^+$ (mg/L) | TN (mg/L) | TP (mg/L) |
|:---:|:---:|:---:|:---:|
| $50.00 \pm 2.40$ | $10.69 \pm 0.27$ | $13.22 \pm 0.51$ | $2.26 \pm 0.19$ |

*2.2. Physicochemical Analysis*

Surface water samples were collected between 9:00 and 9:30 a.m. on Days 0, 7, 14, 21, and 28 of the experiment using 500 mL polyethylene bottles, with three replicates collected at each sampling time. The collected water samples were stored in the dark at 4 °C and analyzed within 48 h. Prior to chemical analysis, water samples were passed through commercial single-use membrane filters (pore diameter of 0.45 mm, Millipore, Burlington, MA, USA). To monitor changes in water quality during the remediation process, COD, NH$_4^+$-N, TN, and TP were analyzed using standard methods, which were the potassium dichromate method, potassium persulfate oxidation–UV spectrophotometric method, Nessler's reagent spectrophotometric method, and the molybdenum–antimony anti-spectrophotometric method, respectively. All analyses were conducted in triplicate, and the average of the three replicate experiments were calculated and used for data interpretation.

*2.3. Microbial Community Analysis*

For the LNM, LNH, and LNV groups, 250 mL of water samples were collected on Days 7, 14, 21, and 28 of the experiment for microbial community structure analysis. Each water sample was filtered through a 0.22 μm filter membrane and stored at −80 °C immediately. To extract DNA, the filter membrane was cut into pieces using sterilized stainless-steel scissors. Microbial community genomic DNA was extracted from the filter membrane samples using the E.Z.N.A.® soil DNA Kit (Omega Bio-tek, Norcross, GA, USA). The 16S rRNA gene was amplified by PCR using primers 338F (5′-ACTCCTACGGGAGGCAGCAG-3′) and 806R (5′-GGACTACHVGGGTWTCTAAT-3′), following the method described in previous studies [23,24], and the PCR products were purified using the AxyPrep DNA Gel Extraction Kit (Axygen Biosciences, Union City, CA, USA). Illumina high-throughput sequencing was conducted by a gene sequencing company (Majorbio, Shanghai, China) to analyze the microbial community diversity and structure of each water sample.

**3. Results and Discussion**

*3.1. Performances of Pollutant Removal*

Overall, the experimental groups (LNM, LNH, and LNV groups) showed significantly higher removal rates of pollutants (COD, NH$_4^+$-N, TN, and TP) than the control group (Figures 2–5). In the early stage of the experiment (0–7 d), there were no significant differences between the experimental and control groups in removal rates of NH$_4^+$-N and TP. On Day 7, the removal rates of NH$_4^+$-N for the experimental groups and control groups were 28% and 22%, respectively, while the removal rates of TP were 19% and 21%, respectively. However, as the plants grew, obvious differences were observed in pollutant removal rates between the experimental and control groups in the later stage of the experiment (7–28 d). In particular, the removal rate of NH$_4^+$-N in the experimental groups was nearly double that of the control group. This might be due to the removal of NH$_4^+$-N in polluted water mainly caused by the processes of aquatic plant absorption and utilization, NH$_4^+$-N volatilization, and microbial nitrification. The abundant leaf surfaces and root systems of aquatic plants provided favorable conditions for microbial growth, thereby promoting the degradation of pollutants in water. On the other hand, the growth of plant roots, especially those of floating plants, in the water enabled direct absorption of nutrients from the water for the plant's own growth [11,25].

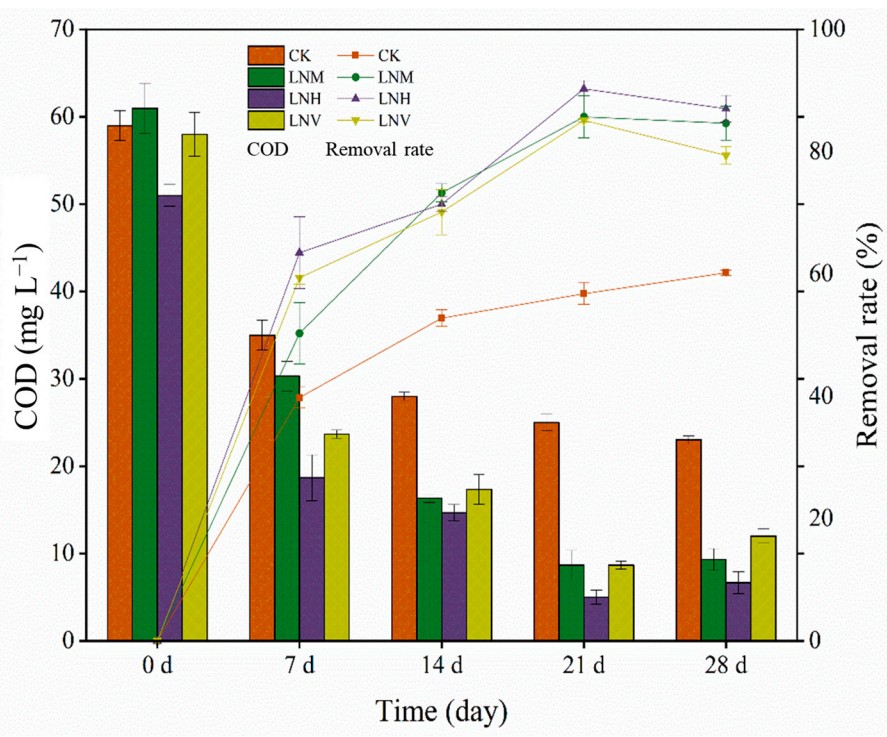

**Figure 2.** COD removal in the LNM, LNH, LNV, and CK systems during the operation process.

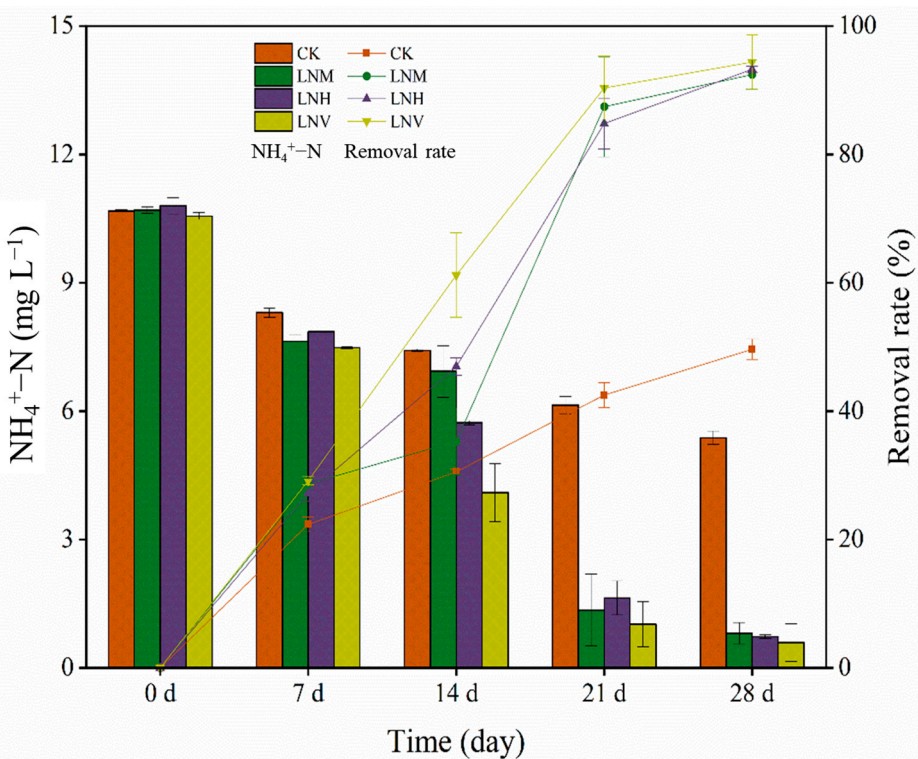

**Figure 3.** $NH_4^+$-N removal in the LNM, LNH, LNV, and CK systems during the operation process.

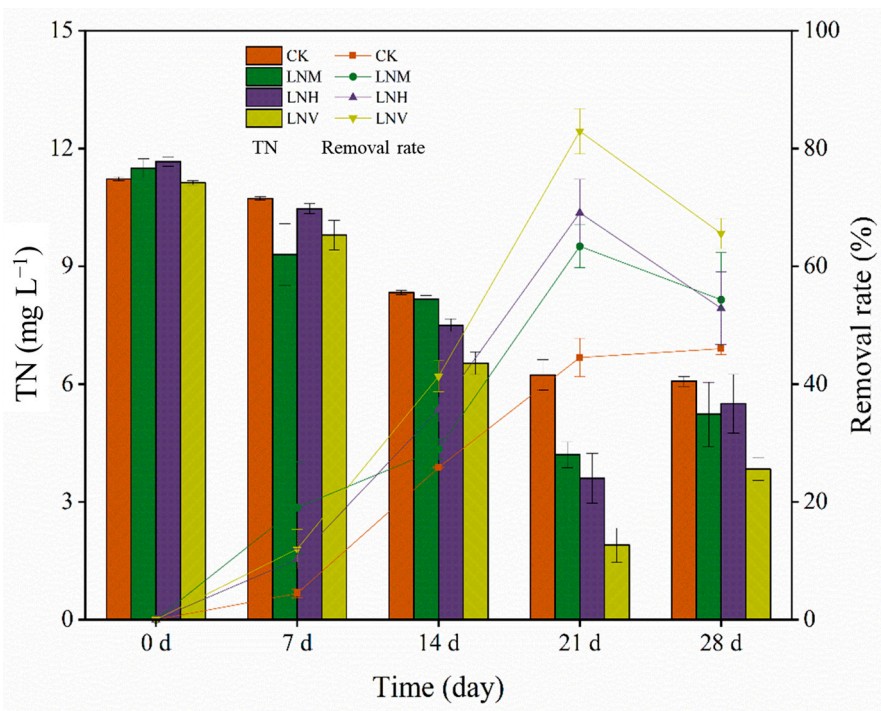

**Figure 4.** TN removal in the LNM, LNH, LNV, and CK systems during the operation process.

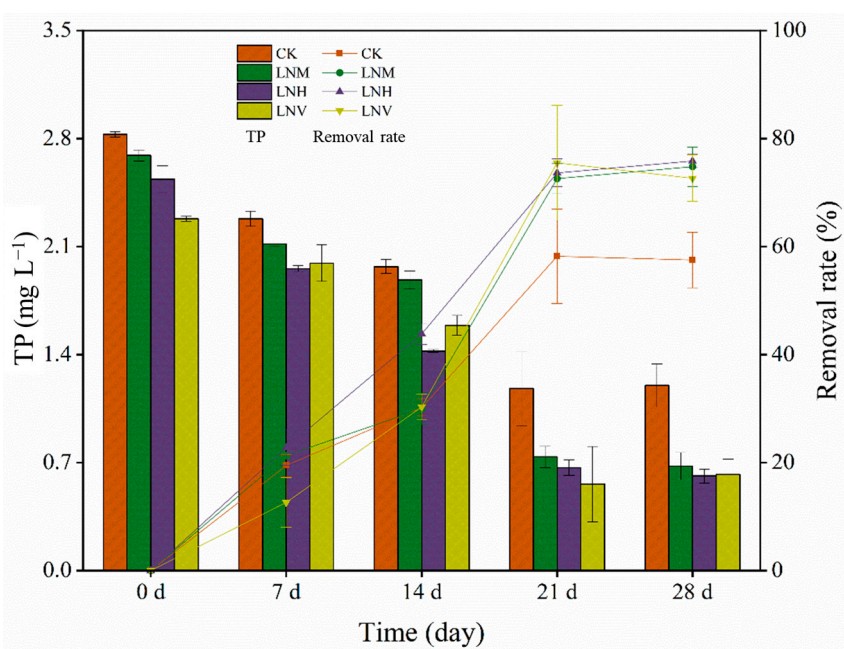

**Figure 5.** TP removal in the LNM, LNH, LNV, and CK systems during the operation process.

### 3.1.1. COD Removal

In the initial stage of the experiment (0–7 d), the water COD concentration decreased rapidly (Figure 2). On Day 7, the COD removal rates of the experimental groups were ranked as LNH (63.49%) > LNV (59.39%) > LNM (50.31%). The COD removal rate decreased in the later stage of the experiment. On Day 21, the maximum COD removal rates were achieved, with LNH (90.29%) > LNV (85.72%) > LNM (85.15%). Similar conclusions have been reported in previous studies [26]. This indicated that the absorption process of plants for organic matter in water can be divided into two stages: a rapid absorption stage in the early stage of the experiment and a slow absorption stage in the later stage. In the early stage of the experiment, the absorption and adsorption of pollutants by aquatic

plants occurred simultaneously, which played a key role in pollutant removal. And the DO (approximately 6.0–7.0 mg/L) in water was sufficient for aerobic microbial degradation and transformation of large organic molecules into small ones, which were quickly absorbed by plants for their own metabolic processes. This process resulted in a significant decrease in COD concentration. In the later stage of the experiment, the absorption and adsorption process of aquatic plants gradually became saturated, and with the consumption of DO (remaining concentration 1.5–2.0 mg/L) in water, the activity of aerobic microorganisms was inhibited, resulting in a decrease in COD removal rate. In addition, on Day 28, the COD concentration in the experimental group showed a slight rebound compared with Day 21, which may be due to an increase in the mortality rate of algae in water, releasing a large amount of dissolved organic matter into the water. Overall, the LNH group showed the highest COD removal efficiency throughout the entire experiment. This might result from the abundant root system of Hydrilla verticillata, which provided a suitable living environment for aerobic organisms to degrade organics to be absorbed and utilized by plants [27].

### 3.1.2. $NH_4^+$-N and TN Removal

The experimental groups showed higher removal rates than the control group (Figure 3). On Day 28, the $NH_4^+$-N removal rates for all experimental groups reached 92%, and there was no significant difference between three experimental groups. The removal of ammonium nitrogen in the polluted water mainly occurred through processes such as absorption and utilization by aquatic plants, ammonium nitrogen volatilization, and microbial nitrification and denitrification. During photosynthesis, plants consumed dissolved $CO_2$ in the water, producing a weak alkaline environment which promoted ammonium nitrogen volatilization [28]. Additionally, plants transported oxygen to the surrounding roots and promoted the degradation of ammonium nitrogen by nitrifying bacteria, resulting in a higher $NH_4^+$-N removal rate. It was worth noticing that in the early stage of the experiment (0–14 d), the $NH_4^+$-N removal rate in the LNV group was significantly higher than the other two experimental groups, which was LNV (61.20%) > LNH (46.95%) > LNM (35.28%). The main reason may be that the wider leaves of Vallisneria provided a better living environment and attachment site for microorganisms, resulting in a higher $NH_4^+$-N removal rate.

The LNV group showed a higher TN removal rate throughout the experiment period with a peak on Day 21, which was LNV (82.94%) > LNH (69.10%) > LNM (63.40%) (Figure 4). The TN removal in the water mainly relied on the aerobic–anaerobic environment of the plant rhizosphere, which provided a favorable environment for the growth of nitrifying and denitrifying bacteria [29]. Studies have reported that plant root exudates not only serve as a carbon source for heterotrophic denitrifying bacteria but also enhance the density of anaerobic ammonia-oxidizing bacteria. Moreover, there was a certain correlation between TN and COD removal rates, as the limited carbon source in the water increased the competition among heterotrophic microorganisms, affecting the nitrification and denitrification processes. Especially, the highest COD removal rate of the LNH group resulted in the lowest C/N ratio in the water, which limited the denitrification process and reduced the TN removal rate [30]. Additionally, on Day 28, the TN concentration increased for all the experimental groups, leading to a decrease in TN removal rate. The same trend was also observed in COD removal rate. In the later stage of the experiment, the partial decay of some plant leaves and increases in algal mortality rate might be a possible explanation.

### 3.1.3. TP Removal

A similar trend of TP removal rate was observed for all three experimental groups (Figure 5). The TP removal rate gradually increased over time and then stabilized. In the early stage of the experiment (0–14 d), significant differences were observed in TP removal rates among three experimental groups. On Day 14, the LNH group showed the highest TP removal rate (43.83%), while those of the LNM and LNV groups were both around 30%. In

the later stage of the experiment (14–28 d), the three experimental groups reached similar TP removal rates. On Day 21, the TP removal rate of the LNV group reached its peak at 75.51%, and on Day 28, the TP removal rates of the LNH and LNM groups reached their peaks at 75.84% and 74.80%, respectively. The removal mechanisms of phosphorus in polluted water included adsorption, precipitation, plant uptake, and plant-mediated microbial processes [31]. It can be seen that the presence of plants in the experimental groups enhanced plant-mediated microbial processes, thereby reducing the TP concentration in the water. Microbes played an important role in phosphorus removal as mineralizers of organic phosphorus via biological mineralization and biochemical mineralization [32]. The highest TP removal rate of the LNH group in the early stage of the experiment may be due to the higher growth rate of Hydrilla verticillata at the beginning of the experiment, which increased the absorption of phosphorus in the water by plants [33].

### 3.2. Microbial Communities Analysis

#### 3.2.1. Microbial Community Structure

As shown in Table 2, alpha diversity was well captured. The Chao index in the late period of the experiment (14–28 d) was significantly higher than that of the early period (0–7 d), indicating that the microbial richness and diversity were higher in the late period relative to the early period. The richness of microbial species of the LNM, LNH, and LNV groups increased from 853, 834, and 893 to 1294, 1243, and 1246. The highest Chao index was observed in the LNM group on Day 14, demonstrating its relatively greater microbial richness and diversity. The Shannon index and Simpson index were indicators for both uniformity and diversity of microbial species in the samples. The change trend of the Shannon index in the LNM group was similar to that of the Chao index, which increased from 2.89 (Day 7) to 4.58 (Day 28). While the Shannon index in the tLNH and LNV groups remained relatively stable in the whole experiment period. The uniformity and diversity of microbial species the in LNM and LNH groups became increasingly bad with time, while that of the LNV group became gradually better on Day 28. This indicated that the plant combination of the LNV group might promote the growth of microorganisms. The difference between the Chao index, Shannon index, and Simpson index might be caused by different secretions and organic matters around different plant roots.

**Table 2.** Diversity index of the samples.

| | Samples | Shannon | Simpson | Chao |
|---|---|---|---|---|
| LNM | LNM7 | 2.89 | 0.1935 | 853 |
| | LNM14 | 4.31 | 0.0608 | 1472 |
| | LNM21 | 4.59 | 0.0202 | 941 |
| | LNM28 | 4.58 | 0.0222 | 1294 |
| LNH | LNH7 | 4.27 | 0.0353 | 832 |
| | LNH14 | 3.90 | 0.0575 | 1185 |
| | LNH21 | 4.55 | 0.0267 | 1035 |
| | LNH28 | 4.40 | 0.0312 | 1300 |
| LNV | LNV7 | 3.75 | 0.0738 | 893 |
| | LNV14 | 4.06 | 0.0487 | 1136 |
| | LNV21 | 4.67 | 0.0254 | 1165 |
| | LNV28 | 3.81 | 0.0983 | 1235 |

#### 3.2.2. Dominant Community at Phylum Level

Previous studies have focused on particle-attached or plant rhizosphere bacterial communities and concluded that bacterial communities were significant contributors to pollutant removal [34]. A total of 13 prokaryotic phyla were identified as the top 1% phyla in all experimental groups (Figure 6). Among them, Proteobacteria (46–88.0%), Actinobacteria (1.7–24.0%), and Bacteroidetes (2.8–17.0%) were the dominant phyla in all experimental groups. However, significant differences were observed in the composition of species in

different experimental groups. The proportion of Proteobacteria in the LNM group showed a gradually decreasing trend with time. On Day 7, the proportion of Proteobacteria was the highest at 88.0% and decreased to 46% on Day 28. The proportion of Proteobacteria in both the LNH group and LNV group showed a trend of first increasing and then decreasing with time, with peaks of 77% and 84% on Day 14. And the lowest proportions were observed on Day 28, which were 50% and 58%, respectively. Proteobacteria was reported to be the most dominant phyla in contaminated water during the bioremediation process. The high abundance of Proteobacteria may accelerate the degradation of low molecular organic matter. The decrease in COD and TN removal rate in all the experimental groups in the later stages of the experiment may also be related to the decrease in the proportion of Proteobacteria [20].

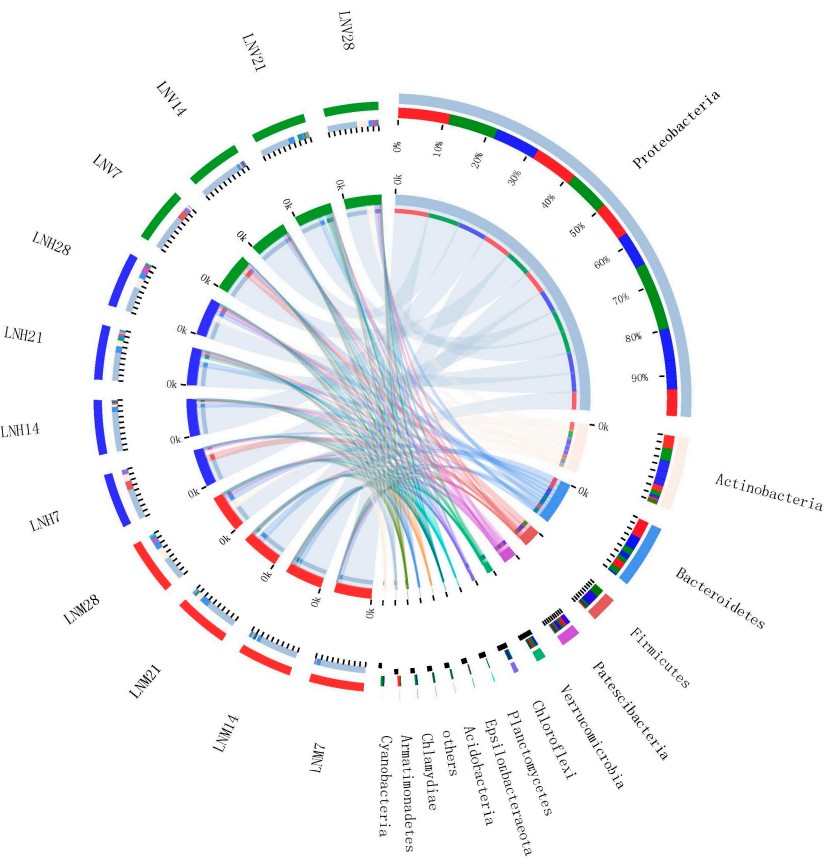

**Figure 6.** The relationship between the relative abundance of dominant phyla and samples.

The proportion of Actinobacteria in both the LNM and LNV groups showed a gradual increase over time, with the lowest values (1.7% for LNM and 2.5% for LNV) appearing on Day 7, and the highest values (24% for LNM and 23% for LNV) appearing on Day 28. However, the proportion of Actinobacteria in the LNH group showed a trend of initially decreasing and then increasing over time, with values of 12% on Day 7, 3.9% on Day 14, and 19% on Day 28. Actinobacteria was reported to be involved in organic matter utilization or degradation for purifying eutrophic wastewater and positively correlated with COD removal [35]. The higher COD removal rate in the LNH group than the LNM and LNV groups during the initial stage of the experiment (0–7 d) may be related to the higher proportion of Actinobacteria.

Although the change trends of the proportion of Bacteroidetes differed from each other for the LNM, LNV, and LNH groups, all experimental groups reached their highest values on Day 21, with LNM (17%) > LNH (12%) > LNV (11%). Although the change trends in Actinobacteria and Bacteroidetes differed among the experimental groups, they all reached their peak values in the later stages of the experiment. This may be attributed to

the depletion of nutrients in the later stages of the experiment, which became a key factor affecting the structure of the microbial community. In addition, through analysis of the microbial community structure throughout the experimental period (0–28 d), we found that on Day 7, the proportion of Firmicute in the LNH and LNV groups reached a maximum value of 32% in both. However, at that time, the proportion of Firmicute in the LNM group was only 2.0%. The COD removal efficiency at this time showed the following trend: LNH (63.49%) > LNV (59.39%) > LNM (50.31%). On Day 14, the proportions of Firmicute in the LNH and LNV groups were 6.9% and 9.5%, respectively, while in the LNM group, it was only 2.5%. The $NH_4^+$-N removal efficiency at this time showed the following trend: LNV (61.20%) > LNH (46.95%) > LNM (35.28%). On Day 21, the proportion of Firmicute in the LNH and LNV groups was 3.6% and 2.1%, respectively, whereas in the LNM group, it was only 0.53%. The TN removal efficiency for the LNH, LNV, and LNM groups showed the following trend: LNV (82.94%) > LNH (69.10%) > LNM (63.40%). Firmicute were major degraders of various organic matter in the decomposition process. We can infer that the higher proportion of Firmicute in the LNH and LNV groups may be related to higher COD, $NH_4^+$-N, and TN removal rates [36].

Similar patterns were also observed for Verrucomicrobi, Chloroflex, and Acidobacteria. The proportion of Verrucomicrobi, Chloroflex, and Acidobacteria in the LNH and LNV groups reached their highest values on Day 21, with percentages of 6.4% for Verrucomicrobi in LNH and 6.0% in LNV, 6.4% for Chloroflex in LNH and 3.9% in LNV, and 2.0% for Acidobacteria in LNH and 2.3% in LNV. However, the proportions of Verrucomicrobi, Chloroflex, and Acidobacteria in the LNM group remained low throughout the experimental period, ranging from 0.1–2.7%, 0.29–1.4%, and 0.14–0.92%, respectively. Both Chloroflex and Verrucomicrobi showed a significant positive correlation with TN removal [37]. Chloroflexi also played a key role in $NH_4^+$-N removal and were reported to be predominant in eutrophic water for biological remediation [38,39]. Acidobacteria were important contributors to the nitrification of nitrogen [40]. These might all contribute to the higher $NH_4^+$-N and TN removal rates of the LNH and LNV groups than the LNM group. Additionally, the proportion of TN-degrading bacteria, such as Chloroflex and Verrucomicrobi, showed a significant decline on Day 28, which led to the decrease in TN removal rates on Day 28.

### 3.2.3. Dominant Community at Class Level

During the initial stage of the experiment (0–14 d), Gammaproteobacteria showed the highest proportion in all experimental groups, with LNV (16%) > LNM (14%) = LNH (14%) (Figure 7). In the later stage (14–28 d), the dominant class changed from Gammaproteobacteria to Alphaproteobacteria, with LNV (14%) > LNM (9.4%) > LNH (8.7%). It can be seen that the proportion of Gammaproteobacteria and Alphaproteobacteria in the LNV group was higher than that in the LNM and LNH groups throughout the whole experiment period. And the proportions of Gammaproteobacteria and Alphaproteobacteria in the LNM and LNH groups were relatively similar. In addition, on Day 7, the proportion of Clostridia in LNV group was significantly higher than that in the LNH and LNM groups, with LNV (15.8%) > LNH (5.4%) > LNM (0.7%). Alphaproteobacteria, Gammaproteobacteria, and Clostridia were positively correlated with nutrient removal. The higher proportion of Alphaproteobacteria, Gammaproteobacteria, and Clostridia in the LNV group than LNM and LNH groups may be attributed to the significantly higher $NH_4^+$-N and TN removal rates. The divergence in bacterial communities among the three experimental groups was mainly due to the influence of different plants and exogenous microbes associated with these plants.

Furthermore, we also noticed that on the Day 7, the proportion of Bacilli in the LNH group reached 10.3%, while that in the LNV and LNM groups was less than 1%. Some species of Bacilli were reported to be microcystin-degrading bacteria [41]. The higher proportion of Bacilli in the LNH group during the initial stage of the experiment may be related to the higher COD removal rate of the LNV and LNM groups.

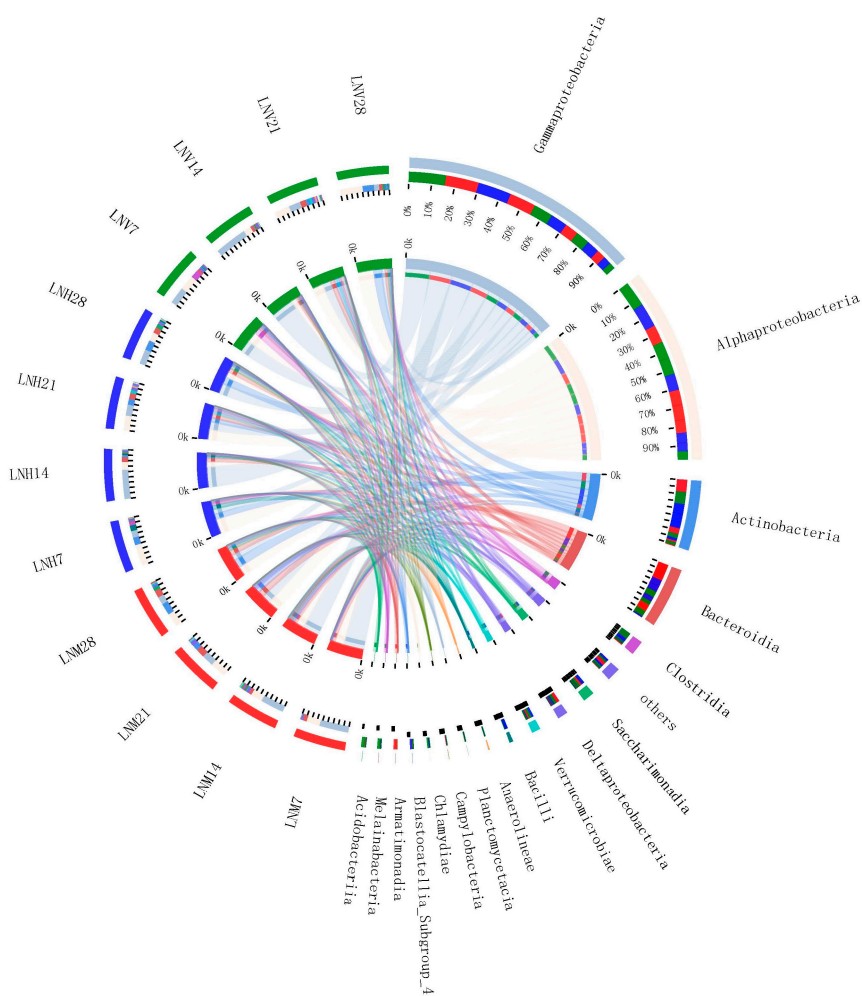

**Figure 7.** The relationship between the relative abundance of dominant class and samples.

### 3.2.4. Dominant Community at Genus Level

In the early stage of the experiment, the most dominant genus in the LNM and LNV groups was *C39*, accounting for 41.6% and 19.2%, respectively, while that of the LNH group was *Rhodobacter*, accounting for 11.2% (Figure 8). In the early stage of the experiment, nutrients were abundant in the water, and the metabolites secreted by plants may be more positively correlated with the shaping of the communities than nutrients. On Day 21, the proportion of *Novosphingobium* increased significantly in the LNV and LNH groups, accounting for 8.2% and 10.3%, respectively. In contrast, the proportion of *Novosphingobium* in the LNM group was the highest on Day 7, accounting for 7.6%, and then showed a decreasing trend during the experiment. *Novosphingobium* are known to be positive degraders of COD [42], which the higher COD removal of the LNH group on Day 21 compared with the LNV and LNM groups may be attributed to. Similarly, the proportion of *Hydrogenophaga* was higher in the LNV and LNH groups on Day 14, accounting for 17.8% and 14.7%, respectively, while that of the LNM group was only 6.1%. *Hydrogenophaga* are known to have a strong positive correlation with $NH_4^+$-N removal, which was consistent with the higher $NH_4^+$-N removal rate of the LNV and LNH groups than that of the LNM group in the early stage of the. Additionally, some *Hydrogenophaga* species might degrade aromatic compounds [43]. On Day 7, the proportion of *Exiguobacterium* was higher in the LNH group, reaching 10.0%, while that of the LNM and LNV groups were lower than 0.1%. *Exiguobacterium* showed a strong positive correlation with TP removal, which led to the higher TP removal rate (43.83%) of the LNH group in the early stage of the experiment compared with the LNM and LNV groups (~30%).

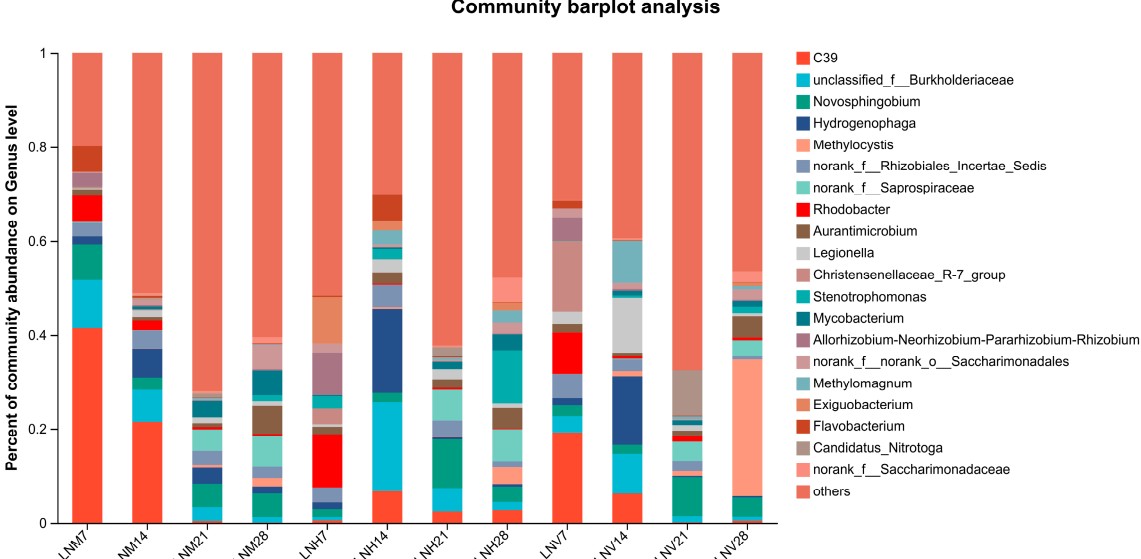

**Figure 8.** Relative abundance of dominant phylogenetic groups at genus level in water phases derived from all treatment microcosms during the operation process.

## 4. Conclusions

In summary, significant differences in COD, $NH_4^+$-N, and TN removal rates among different combined plant systems were observed in this study. The LNH group showed the highest COD removal rate throughout the experiment with a peak of 90.29%, while the LNV group showed more prominent advantages in $NH_4^+$-N and TN removal, with peaks of 92.0% and 82.94%. Although the LNH group had a higher TP removal rate in the early phase, there was no significant difference in TP removal rate among teh three groups at the end of the experiment, suggesting that the combination of plants had little effect on TP removal. In addition, different combined plant systems affected the microbial community structure in water, which led to different pollutant removal efficiencies. The higher abundance of Proteobacteria and Actinobacteria in the LNH group might contribute to higher COD removal, while the higher abundance of $NH_4^+$-N and TN degraders (i.e., Verrucomicrobia, Chloroflexi, and Acidobacteria) in the LNV group led to higher $NH_4^+$-N and TN removal rates. Although no single combination of aquatic plants demonstrated optimal removal rates for nitrogen, phosphorus, and COD, this study has shown that each plant combination has its advantages in specific pollutant removal aspects. In practical applications, apart from removal rate indicators, factors such as availability and adaptability of the plants, as well as variations in nutrient pollution levels, should also be considered to make the best selection of aquatic plant combinations based on specific requirements and conditions. Therefore, appropriate combinations of plants should be selected for the removal of different pollutants, and increasing the abundance of functional microbial communities can further improve the effectiveness of phytoremediation.

**Author Contributions:** Conceptualization, J.J. and S.P.; methodology, H.X.; validation, J.J. and S.P.; formal analysis, H.X.; investigation, J.J.; resources, K.Z.; data curation, K.Z.; writing—original draft preparation, J.J.; writing—review and editing, S.P.; visualization, H.X.; supervision, S.P.; project administration, J.J.; funding acquisition, S.P. All authors have read and agreed to the published version of the manuscript.

**Funding:** This research was funded by National Key Research and Development Program of China, grant number 2022YFC3203400 and The Fundamental Research Funds for the Central Public Welfare Research Institutes, grant number TKS20230302.

**Data Availability Statement:** Not applicable.

**Conflicts of Interest:** The authors declare no conflict of interest. The funders had no role in the design of the study; in the collection, analyses, or interpretation of data; in the writing of the manuscript; or in the decision to publish the results.

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
