# Peer review of "Study on Purification Efficiency of Novel Aquatic Plant Combinations and Characteristics of Microbial Community Disturbance in Eutrophic Water Bodies"

_water, doi:10.3390/w15142586_

Round 1

Reviewer 1 Report

The manuscript discusses interesting experiments on the removal of nutrients and organic matter from polluted waters with the help of aquatic plants. This is a continuation of earlier research, extended to other plant species and analysis of the composition of bacteria. The tests were carried out correctly and their results were presented in a clear way. While reading the manuscript, I found many shortcomings which are presented below.

The Latin names of genera and species should be written in italics, while the names of higher taxonomic units (families, classes and phyla) should be written in normal font. This should be corrected throughout the manuscript.

When citing in the text, we do not give the first name, but only the surname. So line 55 should read Xu et al. instead of Xing-Jian Xu et al. and in line 75 should be Cheng et al. instead of Rui Cheng et al., and so on.

English species names are lowercase, so it should be giant reed instead of Giant reed on line 77.

Previous studies (Xiao et al. 2021) used Lythrum salicaria. Meanwhile, the family name Lythraceae is now used. Was the species changed and the authors had difficulties in identifying it, or did they use a mixture of several species from the Lythraceae family? This should be written in Materials and Methods. This is important, because if there were several species, there could be different effects of individual species on nutrient concentrations.

Why are the species names of Myriophyllum and Vallisneria not given?

In line 104, it should probably be Myriophyllum instead of Nymphaea, because Nymphaea is not a submerged species.

Line 114. This sieve has only 20 mesh? What diameter were these mesh?

Line 134. Not Physio, but Physico.

Line 200. If LNH, it is not Myriophyllum but Hydrilla.

Line 209. If NH4 was analyzed, it should be ammonium, not ammonia.

Lines 266-267. There is: Chao and index. And is redundant.

Line 268. There is Shannnon, should be Shannon.

Lines 280, 337 and 360. There is no need to capitalize Phylum, Class and Genus. Needs to be converted to lowercase.

Line 283. Bacteria belong to the prokaryotic phyla, not eukaryotic.

Line 291. Not pecks, but peaks.

What does Circos mean in Figures 6 and 7?

In Reference No. 13 should be 2021 not 2020.

In Reference 22 should be Hydrobiologia 2011, 674, 133–156.

In Reference No. 27 authors should be written as follows: Machat, H.; Boudokhane, C.; Roche, N.; Dhaouadi, H.

Similarly, you need to change the names of the authors in item 28.

In Reference No. 33 authors should be written as follows: Herlemann, D.P.R.; Manecki, M.; Meeske, C.; Pollehne, F.; Labrenz, M.; Schulz-Bull, D.; Dittmar, T.; Jürgens, K.

Author Response

The manuscript discusses interesting experiments on the removal of nutrients and organic matter from polluted waters with the help of aquatic plants. This is a continuation of earlier research, extended to other plant species and analysis of the composition of bacteria. The tests were carried out correctly and their results were presented in a clear way. While reading the manuscript, I found many shortcomings which are presented below.

RE: Special thanks to you for your good comments!

The Latin names of genera and species should be written in italics, while the names of higher taxonomic units (families, classes and phyla) should be written in normal font. This should be corrected throughout the manuscript.

RE: We carefully examined the relevant errors in the original manuscript. In the new manuscript, the names of higher taxonomic units (families, classes, and phyla) in the abstract, introduction, 3.2.2. Dominant community at phylum level, 3.2.3. Dominant community at class level, and the conclusion section have been corrected.

When citing in the text, we do not give the first name, but only the surname. So line 55 should read Xu et al. instead of Xing-Jian Xu et al. and in line 75 should be Cheng et al. instead of Rui Cheng et al., and so on.

RE: We carefully examined the relevant errors in the manuscript. In the new manuscript, the citation format has been corrected. Line 58: Xu et al.; Line 76: Zuo et al.; Line 79: Cheng et al.

English species names are lowercase, so it should be giant reed instead of Giant reed on line 77.

RE:The incorrect writing of English species names has been corrected in line 80 of the new manuscript.

Previous studies (Xiao et al. 2021) used Lythrum salicaria. Meanwhile, the family name Lythraceae is now used. Was the species changed and the authors had difficulties in identifying it, or did they use a mixture of several species from the Lythraceae family? This should be written in Materials and Methods. This is important, because if there were several species, there could be different effects of individual species on nutrient concentrations.

RE:The study (Xiao et al., 2021) also served as preliminary research for the research team of this paper, thus the species Lythrum salicaria L from the family Lythraceae was still used in this study. In order to enhance the accuracy of the description, it is described in the new manuscript as follows: 'The species names of Lythraceae, Myriophyllum, and Vallisneria were Lythrum salicaria L, Myriophyllum spicatum L, and Vallisneria natans (Lour.) Hara in this study, respectively,' in lines 113-115.

Why are the species names of Myriophyllum and Vallisneria not given?

RE:The specific species of Myriophyllum has been given as Myriophyllum spicatum L. And the specific species of Vallisneria has been given as Vallisneria natans (Lour.) Hara. In order to enhance the accuracy of the description, it is described in the new manuscript as follows: 'The species names of Lythraceae, Myriophyllum, and Vallisneria were Lythrum salicaria L, Myriophyllum spicatum L, and Vallisneria natans (Lour.) Hara in this study, respectively,' in lines 113-115.

In line 104, it should probably be Myriophyllum instead of Nymphaea, because Nymphaea is not a submerged species.

RE:We apologize for the incorrect writing in the original manuscript. In the new manuscript, the sentence in Line 110 has been modified to "which was Myriophyllum, Hydrilla verticillata, and Vallisneria, respectively."

Line 114. This sieve has only 20 mesh? What diameter were these mesh?

RE:The parameter of 20 mesh was used to designate the sieve aperture size, indicating that there are 20 openings per square inch of the mesh, resulting in an approximate diameter of 0.85mm for each opening. In the new manuscript, the description "sieve (20 mesh, diameter 0.85mm)" has been added in line 125.

Line 134. Not Physio, but Physico.

RE:We apologize for the incorrect writing in the original manuscript. The relevant errors have been corrected in line 144 of the new manuscript.

Line 200. If LNH, it is not Myriophyllum but Hydrilla.

RE:We apologize for the incorrect writing in the original manuscript. The relevant errors have been corrected in line 212 of the new manuscript.

Line 209. If NH4 was analyzed, it should be ammonium, not ammonia.

RE:We apologize for the incorrect writing in the original manuscript. The relevant errors have been corrected in line 221, 222, 224, 226 of the new manuscript.

Lines 266-267. There is: Chao and index. And is redundant.

RE:We apologize for the incorrect writing in the original manuscript. In the new manuscript, the word "and" has been removed in line 280.

Line 268. There is Shannnon, should be Shannon.

RE:We apologize for the incorrect writing in the original manuscript. The relevant errors have been corrected in line 282, 284, 285, 290 of the new manuscript.

Lines 280, 337 and 360. There is no need to capitalize Phylum, Class and Genus. Needs to be converted to lowercase.

RE:We apologize for the incorrect writing in the original manuscript. The relevant errors have been corrected in line 294, 360, 384 of the new manuscript.

Line 283. Bacteria belong to the prokaryotic phyla, not eukaryotic.

RE:We apologize for the incorrect writing in the original manuscript. The relevant errors have been corrected in line 297 of the new manuscript.

Line 291. Not pecks, but peaks.

RE:We apologize for the incorrect writing in the original manuscript. The relevant errors have been corrected in line 305 of the new manuscript.

What does Circos mean in Figures 6 and 7?

RE:The word “Circos” indicating that the figure is a circular diagram and unrelated to the research content. It has been removed in Figures 6 and 7 of the new manuscript.

In Reference No. 13 should be 2021 not 2020.

RE:We apologize for the incorrect writing of reference in the original manuscript. The year of reference No. 14 has been corrected in the new manuscript.

In Reference 22 should be Hydrobiologia 2011, 674, 133–156.

RE:We apologize for the incorrect writing of reference in the original manuscript. Reference No. 19, 25, 28, 29, 30, 31, 37, 38, 41, 43 have been corrected in the new manuscript.

In Reference No. 27 authors should be written as follows: Machat, H.; Boudokhane, C.; Roche, N.; Dhaouadi, H.

RE:We apologize for the incorrect writing of reference in the original manuscript. Reference No. 30 has been corrected in the new manuscript.

Similarly, you need to change the names of the authors in item 28.

RE:We apologize for the incorrect writing of reference in the original manuscript. Reference No. 31 has been corrected in the new manuscript.

In Reference No. 33 authors should be written as follows: Herlemann, D.P.R.; Manecki, M.; Meeske, C.; Pollehne, F.; Labrenz, M.; Schulz-Bull, D.; Dittmar, T.; Jürgens, K.

RE:We apologize for the incorrect writing of reference in the original manuscript. Reference No. 36 has been corrected in the new manuscript.

Reviewer 2 Report

This paper summarized the different performances of purification in eutrophic water under three different plant combinations. The research was innovative and provided a theoretical basis for the purification in eutrophic water. Overall, the structure of the entire paper was complete and the experiment was relatively complete, but there were still some issues that need to be noted.

1. Abstract: Please indicate that the components of the control group (CK) were only sediment and culture water without any plants.

2. Introduction:“In addition, it is widely recognized that microbes also play a significant role in pollutant removal through organic nitrogen degradation, nitrification, denitrification, stabilization, volatilization, and other processes.” Recommend some appropriate literatures for explanatory purposes here.

3. 2.1:Please explain why the LNM group chose the Myriophyllum.

4. 3.1.1: “In the early stage of the experiment, the absorption and adsorption of pollutants by aquatic plants played a key role in pollutants removal” This sentence seems inappropriate, and the word “absorption” has appeared twice.

5. 3.1.1: The paper mentioned the effect of DO on microbial activity. Can you provide the data of DO in the experiment to support this statement?

6. 3.2.1: Since the paper mentioned that the higher proportion of Firmicute in the LNH and LNV groups may be related to the high removal rates of COD, NH-N and TN, how to explain the low proportion of Firmicute in the LNM group? (The three groups were quit close in COD, NH-N and TN removal rates.)

7. 3.2.3: For the functions of the Novosphingobium, please cite the relevant literatures.

8. Conclusion: Generally, the removal rates of nitrogen, phosphorus and COD in the purification of water eutrophication should be considered at the same time, while the experimental results showed that the removal rates of nitrogen, phosphorus and the removal rate of COD can not be a high level at the same time. Does this issue need further discussion?

9. Data and figure: In the paper, there is no clear indication of the left and right axes in Figure 2~5. (It is impossible to know which one of the bar graphs or the linear graphs represents the concentration and which one represents the removal rate.)

10. In addition, some grammatical errors should also be noted and modified.

Needing improvement.

Author Response

This paper summarized the different performances of purification in eutrophic water under three different plant combinations. The research was innovative and provided a theoretical basis for the purification in eutrophic water. Overall, the structure of the entire paper was complete and the experiment was relatively complete, but there were still some issues that need to be noted.

RE: Special thanks to you for your good comments!

  1. Abstract: Please indicate that the components of the control group (CK) were only sediment and culture water without any plants.

RE: We have provided additional clarification in the abstract of the new manuscript, "The components of the CK group were only sediment and culture water without any plants" in lines 16-17.

  1. Introduction:“In addition, it is widely recognized that microbes also play a significant role in pollutant removal through organic nitrogen degradation, nitrification, denitrification, stabilization, volatilization, and other processes.” Recommend some appropriate literatures for explanatory purposes here.

RE: We have added references in line 72 of the new manuscript, “Samsó, R.;  García, J., Bacteria distribution and dynamics in constructed wetlands based on modelling results. Sci.Total Environ. 2013, 461, 430-440. Hoang, H.G.; Thuy, B.T.P.; Lin, C.; Vo, D.-V.N.; Tran, H.-T.; Bahari, M.B.; Le, V.-R.; Vu, C.T., The nitrogen cycle and mitigation strategies for nitrogen loss during organic waste composting: A review. Chemosphere. 2022, 300, 134514.” in lines 473-476.

  1. 2.1:Please explain why the LNM group chose the Myriophyllum.

RE: We have provided additional clarification in the new manuscript, "Among them, it has been demonstrated through research that Myriophyllum, as a submerged aquatic plant, has significant effects on the removal rates of COD, TN, and TP in eutrophic water bodies[13,14]." in lines 111-113.

  1. 3.1.1: “In the early stage of the experiment, the absorption and adsorption of pollutants by aquatic plants played a key role in pollutants removal” This sentence seems inappropriate, and the word “absorption” has appeared twice.

RE: We have provided additional clarification in the new manuscript, "In the early stage of the experiment, the absorption and adsorption of pollutants by aquatic plants occurred simultaneously, which played a key role in pollutants removal." in lines 198-200.

  1. 3.1.1: The paper mentioned the effect of DO on microbial activity. Can you provide the data of DO in the experiment to support this statement?

RE: We have added the relevant data in the new manuscript, “and the DO (approximately 6.0-7.0 mg/L) in water was sufficient for aerobic microbial degradation and transformation of large organic molecules into small ones, which were quickly absorbed by plants for their own metabolic processes.” in line 201, “and with the consumption of DO(remaining concentration 1.5-2.0 mg/L) in water” in line 206.

  1. 3.2.1: Since the paper mentioned that the higher proportion of Firmicute in the LNH and LNV groups may be related to the high removal rates of COD, NH-N and TN, how to explain the low proportion of Firmicute in the LNM group? (The three groups were quit close in COD, NH-N and TN removal rates.)

RE: Through analysis of the microbial community structure throughout the experimental period (0-28d), we found that on day 7, the proportion of Firmicute in the LNH and LNV groups reached a maximum value of 32% both. However, at that time, the proportion of Firmicute in the LNM group was only 2.0%. The COD removal efficiency at this time showed the following trend: LNH (63.49%) > LNV (59.39%) > LNM (50.31%). On day 14, the proportion of Firmicute in the LNH and LNV groups was 6.9% and 9.5%, respectively, while in the LNM group, it was only 2.5%. The NH4+-N removal efficiency at this time showed the following trend: LNV (61.20%) > LNH (46.95%) > LNM (35.28%). On day 21, the proportion of Firmicute in the LNH and LNV groups was 3.6% and 2.1%, respectively, whereas in the LNM group, it was only 0.53%. The TN removal efficiency for the LNH, LNV, and LNM groups showed the following trend: LNV (82.94%) > LNH (69.10%) > LNM (63.40%). Firmicute were major degraders of various organic matters in the decomposition process. We can infer that the higher proportion of Firmicute in LNH and LNV groups may be related to higher COD, NH4+-N, and TN removal rates. We have provided additional explanations in lines 328-339 of the new manuscript.

  1. 3.2.3: For the functions of the Novosphingobium, please cite the relevant literatures.

RE: We have added reference in line 394 of the new manuscript, “Chen, Y.;  Chai, L.;  Tang, C.;  Yang, Z.;  Zheng, Y.;  Shi, Y.;  Zhang, H., Kraft lignin biodegradation by Novosphingobium sp. B-7 and analysis of the degradation process. Bioresour. Technol.  2012, 123, 682-685.” in lines 530-531.

  1. Conclusion: Generally, the removal rates of nitrogen, phosphorus and COD in the purification of water eutrophication should be considered at the same time, while the experimental results showed that the removal rates of nitrogen, phosphorus and the removal rate of COD can not be a high level at the same time. Does this issue need further discussion?

RE: We have provided additional explanations in the conclusion section of the new manuscript,“Although no single combination of aquatic plants demonstrated optimal removal rates for nitrogen, phosphorus, and COD, this study has shown that each plant combination has its advantages in specific pollutant removal aspects. In practical applications, apart from removal rate indicators, factors such as availability and adaptability of the plants, as well as variations in nutrient pollution levels, should also be considered to make the best selection of aquatic plant combinations based on specific requirements and conditions.” in lines 422-428.

  1. Data and figure: In the paper, there is no clear indication of the left and right axes in Figure 2~5. (It is impossible to know which one of the bar graphs or the linear graphs represents the concentration and which one represents the removal rate.)

RE: In the new manuscript, we have added explanatory labels for the left and right axes in the legends of Figures 2~5.

  1. In addition, some grammatical errors should also be noted and modified.

RE:We have made correction according to the reviewer’s comments. The English has been thoroughly improved and the formal deficiencies have been revised carefully.

Reviewer 3 Report

This manuscript is a really nice work and has significant contribution to the study of eutrophication control based on aquatic plant combinations and characteristics of microbial community disturbance. The results presentation is given is clear and generally the manuscript is well written. I believe that this study could become a nice contribution to the Water journal, after some minor modifications:

·        Please add some relevant text regarding the issues caused by eutrophication in the introduction part (like possible anoxia)

·        In the introduction part, give some more emphasis regarding the novelty of your work and how it could contribute to the scientific community

Author Response

This manuscript is a really nice work and has significant contribution to the study of eutrophication control based on aquatic plant combinations and characteristics of microbial community disturbance. The results presentation is given is clear and generally the manuscript is well written. I believe that this study could become a nice contribution to the Water journal, after some minor modifications:

RE: Special thanks to you for your good comments!

  • Please add some relevant text regarding the issues caused by eutrophication in the introduction part (like possible anoxia)

RE: In the introduction section of the new manuscript, we have supplemented the explanation regarding the issues caused by eutrophication, “Eutrophication of water bodies can trigger algal blooms and water blooms, deplete oxygen in the water, cause water turbidity, damage fishery resources, affect drinking water safety, and disrupt ecological balance, among a series of environmental issues.” in lines 35-38. And we have added reference in line 453 of the new manuscript, “Bhateria, R.; Jain, D., Water quality assessment of lake water: a review. Sustain. Water Resour. Manag.  2016, 2, 161-173.”.

  • In the introduction part, give some more emphasis regarding the novelty of your work and how it could contribute to the scientific community

RE: In the introduction section of the new manuscript, we have supplemented the explanation regarding the novelty of this work, “This study investigates the interactions among different aquatic plants to uncover the synergistic effects of various combinations of aquatic plants in the restoration of eutrophicated water bodies, which establishes a comprehensive restoration strategy using a combination of aquatic plants for eutrophication control in water bodies.”in lines 96-100.

Round 2

Reviewer 2 Report

Much better